# Discovery of New 1,4,6-Trisubstituted-1*H*-pyrazolo[3,4-*b*]pyridines with Anti-Tumor Efficacy in Mouse Model of Breast Cancer [note 1]

**DOI:** 10.3390/pharmaceutics15030787

**Published:** 2023-02-27

**Authors:** Maria Georgiou, Nikolaos Lougiakis, Roxane Tenta, Katerina Gioti, Stavroula Baritaki, Lydia-Evangelia Gkaralea, Elisavet Deligianni, Panagiotis Marakos, Nicole Pouli, Dimitris Stellas

**Affiliations:** 1Division of Pharmaceutical Chemistry, Department of Pharmacy, School of Health Sciences, National and Kapodistrian University of Athens, Panepistimiopolis Zografou, 15771 Athens, Greece; 2Department of Nutrition & Dietetics, School of Health Sciences and Education, Harokopio University, 17671 Athens, Greece; 3Laboratory of Experimental Oncology, Division of Surgery, School of Medicine, University of Crete, 71003 Heraklion, Greece; 4Institute of Chemical Biology, National Hellenic Research Foundation, 11635 Athens, Greece

**Keywords:** purine analogues, pyrazolopyridine, antiproliferative activity, anticancer agent, in vivo, breast cancer

## Abstract

Purine analogues are important therapeutic tools due to their affinity to enzymes or receptors that are involved in critical biological processes. In this study, new 1,4,6-trisubstituted pyrazolo[3,4-*b*]pyridines were designed and synthesized, and their cytotoxic potential was been studied. The new derivatives were prepared through suitable arylhydrazines, and upon successive conversion first to aminopyrazoles, they were converted then to 1,6-disubstituted pyrazolo[3,4-*b*]pyridine-4-ones; this served as the starting point for the synthesis of the target compounds. The cytotoxic activity of the derivatives was evaluated against several human and murine cancer cell lines. Substantial structure activity relationships (SARs) could be extracted, mainly concerning the 4-alkylaminoethyl ethers, which showed potent in vitro antiproliferative activity in the low μM level (0.75–4.15 μΜ) without affecting the proliferation of normal cells. The most potent analogues underwent in vivo evaluation and were found to inhibit tumor growth in vivo in an orthotopic breast cancer mouse model. The novel compounds exhibited no systemic toxicity; they affected only the implanted tumors and did not interfere with the immune system of the animals. Our results revealed a very potent novel compound which could be an ideal lead for the discovery of promising anti-tumor agents, and could also be further explored for combination treatments with immunotherapeutic drugs.

## 1. Introduction

Nitrogen containing fused heterocyclic rings that present a structural analogy to adenine are usually considered privileged scaffolds, since they are often found in many biologically active and clinically useful derivatives. Among them, pyrazolopyrimidine and the closely related pyrazolopyridine pharmacophores are present in druggable small molecule entities, marketed drugs, such as Allopurinol, Zaleplon, Sildenafil, Ibrutinib, and compounds such as Etazolate, that undergo clinical evaluation (Figure 1).

Several pyrazolo[3,4-*b*]pyridine-based derivatives that bear diverse substitutions were reported to exhibit potent antiviral [1,2] and antibacterial [3,4] properties; to inhibit important enzymes, such as phosphodiesterase-4 [5], or neutrophil elastase [6]; and to serve as ligands for A1-adenosine [7] or prostaglandin E2 receptor 1 [8]. The anti-cancer potential of this class of compounds is of particular interest as they show antiproliferative activity, apoptosis induction, and angiogenesis inhibition [9,10,11], albeit through diverse molecular targets and mechanisms of action, such as targeting tubulin polymerization [12] and protein kinase signal transduction in cancer cells [10,13,14,15].

Our long-term aims are the discovery of novel purine analogues with in vitro and in vivo cytotoxic activity [16,17]. In this respect, we have previously reported on numerous pyrazolo[3,4-*c*]pyridines with potent antiproliferative activity, bearing suitable substituents at critical positions of the central scaffold [18,19]. We observed that the most interesting compounds possessed a 3-aryl group and were also substituted at positions 5 and/or 7 of the original pyrazolo[3,4-*c*]pyridine nucleus. As a continuation of this effort we have designed, synthesized, and evaluated the cytotoxic activity of a new series of compounds which bear the pyrazolo[3,4-*b*]pyridine ring system as their central core. Based on our earlier findings, we have inserted a variety of substituents in specific sites of this scaffold, taking care to preserve the above mentioned substitution pattern, in order to investigate whether the new structural analogues maintain the cytotoxic activity, thereby assisting in the extraction of helpful structure activity relationships. We have also evaluated the efficacy of our newly synthesized analogues in vitro, and the three most promising among the new compounds were subsequently evaluated in an orthotopic syngeneic mouse model of breast cancer, prompted by our in vitro results.

## 2. Materials and Methods

### 2.1. Synthetic Procedures and Analytical Data

Melting points were determined on a Büchi apparatus and are uncorrected. ^1^H NMR spectra and 2D spectra were recorded on a Bruker Avance III 600 or a Bruker Avance DRX 400 instrument, whereas ^13^C NMR spectra were recorded on a Bruker Avance III 600 spectrometer in deuterated solvents and were referenced to TMS (*δ* scale). The signals of ^1^H and ^13^C spectra were unambiguously assigned by using 2D NMR techniques: ^1^H^1^H COSY, NOESY, HMQC, and HMBC. Mass spectra were recorded with a LTQ Orbitrap Discovery instrument, possessing an Ionmax ionization source. The purity of the key compounds (>95%) was determined on a Thermo Finnigan^®^ HPLC System (P4000 Pump, AS3000 Autosampler, UV Spectra System UV6000LP detector, Chromquest™ 4.1 Software); Fortis^®^ UniverSil HS-C18 (150 mm, 4.6 mm, 5 um); mobile phase 1% acetic acid in water/acetonitrile; flow rate 1 mL/min; column temperature 25 °C; injection volume 10 μL; absorbance at 254 nm). Flash chromatography was performed on Silicagel ACROS ORGANICS 40–60 μm and 60–200 μm, 60A. Analytical thin layer chromatography (TLC) was carried out on precoated (0.25 mm) Merck silica gel F-254 plates.

2,2-Dimethyl-5-{(methylthio)[(1-phenyl-1H-pyrazol-5-yl)amino]methylene}-1,3-dioxane-4,6-dione **(5a)**

Dioxanedione **4** (5.55 g, 22 mmoL) was added to a solution of the amine **3a** (3.38 g, 21 mmoL) [20] in ethanol (50 mL)[20], and the resulting solution was refluxed for 1 h. The solvent was vacuum-evaporated, and the residue was purified by flash column chromatography (silica gel 40–60 μm), and a mixture of cyclohexane/EtOAc (8/2 to 5/5, *v*/*v*) was used as the eluent, resulting in **5a** (5 g, 78%) as a white solid, mp. 144–146 °C (Εt_2_O). ^1^H-NMR (600 MHz, CDCl_3_) δ (ppm) 1.70 (s, 6H, 2xCH_3_), 2.10 (s, 3H, SCH_3_), 6.48 (d, 1H, *J* = 1.8 Hz, pyrazole H-4), 7.36–7.41 (m, 1H, phenyl H-4), 7.46–7.49 (m, 4H, phenyl H-2, H-3, H-5, H-6), 7.71 (d, 1H, *J* = 1.8 Hz, pyrazole H-3), 12.52 (brs, 1H, D_2_O exch., NH). ^13^C NMR (151 MHz, CDCl_3_) δ (ppm) 18.23, 26.47, 88.57, 103.53, 105.17, 124.25, 128.52, 129.46, 135.34, 138.05, 140.44, 163.67, 179.31. HR-MS (ESI) *m/z* calculated for C_17_H_18_N_3_O_4_S [M+H]^+^: 358.0867; found 358.0846.

2,2-Dimethyl-5-{[(methylthio)(1-(3-fluorophenyl)-1H-pyrazol-5-yl)amino]methylene}-1,3-dioxane-4,6-dione **(5b)**

This derivative was prepared by a method analogous to that described for the synthesis of **5a**. Chromatographic purification (silica gel 40–60 μm) was performed by using a mixture of cyclohexane/EtOAc (8/2 to 5/5, *v*/*v*) as the eluent. Yield: 65%. White solid, mp. 153–155 °C (CH_2_Cl_2_/*n*-pentane). ^1^H-NMR (600 MHz, CDCl_3_) δ (ppm) 1.72 (s, 6H, 2xCH_3_), 2.14 (s, 3H, SCH_3_), 6.48 (d, 1H, *J* = 2.0 Hz, pyrazole H-4), 7.09 (m, 1H, 3-fluorophenyl H-4), 7.22–7.29 (m, 2H, 3-fluorophenyl H-2, H-6), 7.42 (m, 1H, 3-fluorophenyl H-5), 7.71 (d, 1H, *J* = 2.0 Hz, pyrazole H-3), 12.50 (brs, 1H, D_2_O exch., NH). ^13^C NMR (151 MHz, CDCl_3_) δ (ppm) 18.35, 26.49, 88.76, 103.72, 105.85, 111.53, 111.70, 115.34, 115.48, 119.42, 119.43, 130.78, 130.84, 135.40, 139.40,139.46, 140.87, 162.09, 163.74, 165.69, 179.40. HR-MS (ESI) *m/z* calculated for C_17_H_15_FN_3_O_4_S [M-H]^−^: 376.0772; found 376.0763.

2,2-Dimethyl-5-{[(1-phenyl-*1H*-pyrazol-5-ylamino)(phenylamino)]methylene}-1,3-dioxane-4,6-dione **(6a)**

Aniline (0.41 mL, 4.5 mmol) was added to a suspension of the methylthio derivative **5a** (1.5 g, 4.18 mmol) in ethanol (17 mL), and the mixture was refluxed for 3 h. The solvent was vacuum-evaporated, and the residue was purified by flash column chromatography (silica gel 40–60 μm), and a mixture of cyclohexane/EtOAc (9/1 to 7/3, *v*/*v*) was used as the eluent, resulting in **6a** (1.1 g, 65 %) as a foam. ^1^HNMR (600 MHz, CDCl_3_) δ (ppm) 1.74 (s, 6H, 2xCH_3_), 5.77 (d, 1H, *J* = 1.7 Hz, pyrazole H-4), 6.77 (d, 2H, *J* = 7.5 Hz, aniline H-2, H-6), 7.09–7.16 (m, 4H, pyrazole H-3, aniline H-3, H-4, H-5), 7.36–7.52 (m, 5H, phenyl H-2, H-3, H-4, H-5, H-6), 11.63 (brs, D_2_O exch., 1H, NH), 12.08 (brs, D_2_O exch., 1H, NH). ^13^C NMR (151 MHz, CDCl_3_) δ (ppm) 26.46, 27.03, 75.37, 103.43, 104.48, 123.65, 124.30, 127.23, 128.24, 128.88, 129.52, 133.64, 135.24, 138.13, 139.78, 160.55, 166.52, 167.24. HR-MS (ESI) *m/z* calculated for C_22_H_21_N_4_O_4_ [M+H]^+^: 405.1557; found 405.1557.

2,2-Dimethyl-5-{[(1-(3-fluorophenyl)-1H-pyrazol-5-ylamino)(phenylamino)]methylene}-1,3-dioxane-4,6-dione **(6b)**

This derivative was prepared by a method analogous to that described for the synthesis of **6a**. Chromatographic purification (silica gel 60–200 μm) was performed by using a mixture of cyclohexane/EtOAc (9/1 to 7/3, *v*/*v*) as the eluent. Yield: 80%. Yellow oil. ^1^HNMR (600 MHz, CDCl_3_) δ (ppm) 1.77 (s, 6H, 2xCH_3_), 5.81 (d, 1H, *J* = 1 Hz, pyrazole H-4), 6.77 (d, 2H, *J* = 7.5 Hz, aniline H-2, H-6), 7.07–7.22 (m, 6H, 3-fluorophenyl H-2, H-4, aniline H-3, H-4, H-5, pyrazole H-3), 7.27 (m, 1H, 3-fluorophenyl H-6), 7.43 (m, 1H, 3-fluorophenyl H-5), 11.67 (brs, D_2_O exch., 1H, NH), 12.10 (brs, D_2_O exch., 1H, NH).^13^C NMR (151 MHz, CDCl_3_) δ (ppm) 26.44, 75.39, 103.55, 105.08, 110.99, 111.15, 114.98, 115.12, 118.66, 118.67, 124.45, 127.42, 128.92, 130.73, 130.79, 133.84, 135.06, 139.45, 139.52, 140.19, 160.76, 162.15, 163.79, 166.50, 167.30. HR-MS (ESI) *m/z* calculated for C_22_H_20_FN_4_O_4_ [M+H]^+^: 423.1463; found 423.1463.

1-Phenyl-6-(phenylamino)-*1H*-pyrazolo[3,4-*b*]pyridin-4-ol **(7a)**

A solution of **6a** (1.5 g, 3.71 mmol) in diphenyl ether (10 mL) was refluxed under argon for 1 h. The reaction mixture was cooled to ambient temperature and purified by flash column chromatography (40–60 silica gel); a mixture of cyclohexane/EtOAc (9/1 to 4.5/5.5, *v*/*v*) was used as the eluent, resulting in **7a** as a white solid. Yield 85%., mp. 117–120 °C (Εt_2_O). ^1^H-NMR (600 MHz, CDCl_3_) δ (ppm) 5.91 (s, 1H, H-5), 6.87 (brs, 1H, D_2_O exch., NH), 6.98 (t, 1H, *J* = 7.1 Hz, phenyl H-4), 7.18–7.21 (m, 3H, phenyl H-3, H-5, anilineH-4), 7.31–7.36 (m, 4H, aniline H-2,H-3,H-5,H-6), 7.94 (d, 2H, *J* = 7.8 Hz, phenyl H-2, H-6), 7.99 (s, 1H, H-3). ^13^C NMR (151 MHz, CDCl_3_) δ (ppm) 90.37, 104.93, 121.02, 122.07, 123.51, 126.57, 129.15, 129.22, 132.87, 138.87, 139.66, 150.87, 157.31, 161.17. HPLC purity at 254 nm, 98.24%. HR-MS (ESI) *m/z* calculated for C_18_H_15_N_4_O [M+H]^+^: 303.1240; found 303.1242.

1-(3-Fluorophenyl)-6-(phenylamino)-*1H*-pyrazolo[3,4-*b*]pyridin-4-ol **(7b)**

This derivative was prepared by a method analogous to that described for the synthesis of **7a**. Chromatographic purification (silica gel 60–200 μm) was performed by using a mixture of cyclohexane/EtOAc (10/1 to 1/10, *v*/*v*) as the eluent. Yield: 85%. mp. 113–115 °C (EtOAc/*n*-pentane). ^1^H-NMR (600 MHz, acetone-*d_6_*) δ (ppm) 6.26 (s, 1H, H-5), 7.01–7.05 (m, 2H, 3-fluorophenyl H-4, aniline H-4), 7.36 (t, 2H, *J* = 7.3 Hz, aniline H-3, H-5), 7.53 (m, 1H, 3-fluorophenyl H-5), 7.79 (d, 2H, *J* = 7.0 Hz, aniline H-2, H-6), 8.10 (s, 1H, H-3), 8.27 (m, 1H, 3-fluorophenyl H-6), 8.39 (m, 1H, 3-fluorophenyl H-2), 10.14 (brs, 1H, D_2_O exch., NH). ^13^C NMR (151 MHz, acetone-*d_6_*) δ (ppm) 91.68, 105.20, 107.74, 107.92, 112.07, 112.21, 116.35, 120.60, 122.73, 129.52, 131.13, 131.20, 133.36, 142.04, 142.68, 142.75, 153.19, 158.71, 160.13, 162.98, 164.59. HPLC purity at 254 nm, 96.29%. HR-MS (ESI) *m/z* calculated for C_18_H_14_FN_4_O [M+H]^+^: 321.1146; found 321.1153.

4-Benzyloxy-*N*,1-diphenyl-*1H*-pyrazolo[3,4-*b*]pyridin-6-amine **(8a)**

K_2_CO_3_ (47 mg, 0.34 mmol) was added to a solution of the pyridinone **7a** (50 mg, 0.17 mmol) in DMF (3.5 mL); the mixture was stirred for 20 min. Then, benzyl bromide (40 μL, 0.34 mmoL) was added, and the resulting mixture was heated at 50 °C for 3 h. It was then poured into ice water, acidified with 9% HCl solution (pH 3–4), and the precipitate was filtered, dried (CaCl_2_) and purified by flash column chromatography (silica gel 40–60 μm); CH_2_Cl_2_ was used as the eluent to provide pure **8a** (50 mg, 65%) as a white solid. Mp. 228–230 °C (ΕtOAc). ^1^H-NMR (600 MHz, CDCl_3_) δ (ppm) 5.22 (s, 2H, benzyloxy CH_2_), 6.09 (s, 1H, H-5), 6.68 (brs,1H, D_2_O exch., NH), 7.10 (t, 1H, *J* = 7.6 Hz, aniline H-4), 7.27 (t, 1H, *J* = 7.3 Hz, phenyl H-4), 7.34 (d, 2H, *J* = 7.0 Hz, aniline H-3, H-5), 7.38–7.45 (m, 7H, benzyloxy 5H, aniline H-2, H-6), 7.49 (t, 2H, *J* = 7.3 Hz, phenyl H-3, H-5), 8.07 (s, 1H, H-3), 8.26 (d, 2H, *J* = 8.0 Hz, phenyl H-2, H-6). ^13^C NMR (151 MHz, CDCl_3_) δ (ppm) 70.48, 87.63, 104.75, 120.90, 121.29, 123.38, 125.74, 127.57, 128.59, 128.94, 128.98, 129.33, 132.38, 135.73, 139.97, 140.18, 151.85, 157.43, 160.83. HPLC purity at 254 nm, 97.89%. HR-MS (ESI) *m/z* calculated for C_25_H_21_N_4_O [M+H]^+^: 393.1710; found 393.1716.

*N*,1-Diphenyl-4-(isopentyloxy)- *1H*-pyrazolo[3,4-*b*]pyridin-6-amine **(8b)**

This derivative was prepared by a method analogous to that described for the synthesis of **8a**. Chromatographic purification (silica gel 40–60 μm) was performed by using CH_2_Cl_2_ as the eluent. Yield: 98%. White solid, mp. 162–164 °C (ΕtOAc/*n*-pentane). ^1^H-NMR (600 MHz, CDCl_3_) δ (ppm) 1.00 (d, 6H, *J* = 6.7 Hz, isopentyl (CH_3_)_2_CH), 1.77 (m, 2H, isopentyl CH_2_CH_2_O), 1.88 (m, 1H, isopentyl -CH), 4.13 (m, 2H, isopentyl CH_2_O), 6.02 (s, 1H, H-5), 6.73 (brs,1H, D_2_O exch., NH), 7.09 (t, 1H, *J* = 7.3 Hz, aniline H-4), 7.27 (t, 1H, *J* = 7.3 Hz, phenyl H-4), 7.36 (t, 2H, *J* = 7.3 Hz, aniline H-3, H-5), 7.49 (t, 2H, *J* = 7.3 Hz, phenyl H-3, H-5), 7.54 (d, 2H, *J* = 7.0 Hz, aniline H-2, H-6), 8.03 (s, 1H, H-3), 8.28 (d, 2H, *J* = 7.0 Hz, phenyl H-2, H-6). ^13^C NMR (151 MHz, CDCl_3_) δ (ppm) 22.66, 25.24, 37.65, 67.17, 87.02, 104.73, 120.71, 121.20, 123.15, 125.61, 128.92, 129.24, 132.32, 140.03, 140.37, 151.85, 157.47, 161.26. HPLC purity at 254 nm, 96.54%. HR-MS (ESI) *m/z* calculated for C_23_H_24_N_4_NaO [M+Na]^+^: 395.1843; found 395.1850.

4-(2-Bromoethoxy)-*N*,1-diphenyl-*1H*-pyrazolo[3,4-*b*]pyridin-6-amine **(8c)**

This derivative was prepared by a method analogous to that described for the synthesis of **8a**. The reaction was completed after 6 h. Chromatographic purification (silica gel 40–60 μm) was performed by using a mixture of cyclohexane/CH_2_Cl_2_/THF (5/5/1 to 4/5/2 *v*/*v*) as the eluent. Yield: 70%. mp. 199–200 °C (ΕtOAc/*n*-pentane). ^1^H-NMR (600 MHz, acetone-*d_6_*) δ (ppm) 3.89 (d, 2H, *J* = 7.0 Hz, bromoethyl C*H_2_*CH_2_O), 4.60 (d, 2H, *J* = 7.0 Hz, bromoethyl CH_2_CH_2_O), 6.30 (s, 1H, H-5), 7.02 (t, 1H, *J* = 7.3 Hz, aniline H-4), 7.30 (t, 1H, *J* = 7.3 Hz, phenyl H-4), 7.35 (t, 2H, *J* = 7.3 Hz, aniline H-3, H-5), 7.54 (t, 2H, *J* = 7.3 Hz, phenyl H-3, H-5), 7.85 (d, 2H, *J* = 7.0 Hz, aniline H-2, H-6), 8.02 (s, 1H, H-3), 8.38 (d, 2H, *J* = 7.0 Hz, phenyl H-2, H-6), 8.62 (brs, 1H, D_2_O exch., NH). ^13^C NMR (151 MHz, acetone-*d_6_*) δ (ppm) 29.89 (overlapping with solvent), 69.24, 89.49, 104.78, 120.32, 121.46, 122.72, 126.24, 129.56, 129.66, 132.57, 141.13, 142.09, 152.61, 158.49, 160.66. HPLC purity at 254 nm, 96.92%. HR-MS (ESI) *m/z* calculated for C_20_H_18_BrN_4_O [M+H]^+^: 409.0659; found 409.0667.

1-(3-Fluorophenyl)-4-benzyloxy-*N*-phenyl-*1H*-pyrazolo[3,4-*b*]pyridin-6-amine **(8d)**

This derivative was prepared by a method analogous to that described for the synthesis of **8a**. Chromatographic purification (silica gel 40–60 μm) was performed by using CH_2_Cl_2_ as the eluent. Yield: 90%. White solid, mp. 198–200 °C (EtOAc). ^1^H-NMR (600 MHz, CDCl_3_) δ (ppm) 5.23 (s, 2H, benzyloxy CH_2_), 6.11 (s, 1H, H-5), 6.67 (brs, 1H, D_2_O exch., NH), 6.95 (t, 1H, *J* = 7.4 Hz, 3-fluorophenyl H-4), 7.12 (t, 1H, *J* = 7.3 Hz, aniline H-4), 7.35–7.44 (m, 10H, benzyloxy 5H, aniline H-2, H-3, H-5, H-6, 3-fluorophenyl H-5), 8.06 (s, 1H, H-3), 8.11 (m, 1H, 3-fluorophenyl H-6), 8.20 (m, 1H, 3-fluorophenyl H-2). ^13^C NMR (151 MHz, CDCl_3_) δ (ppm) 70.51, 87.77, 104.92, 108.11, 108.29, 112.03, 112.17, 116.09, 121.09, 123.62, 127.58, 128.64, 128.96, 129.39, 130.09, 130.15, 132.82, 135.62, 139.96, 141.41, 141.48, 152.11, 157.60, 160.82, 162.31, 163.93. HPLC purity at 254 nm, 99.02%. HR-MS (ESI) *m/z* calculated for C_25_H_20_FN_4_O [M+H]^+^: 411.1616;found 411.1615.

1-(3-Fluorophenyl)-4-(isopentyloxy)-*N*-phenyl-*1H*-pyrazolo[3,4-*b*]pyridin-6-amine **(8e)**

This derivative was prepared by a method analogous to that described for the synthesis of **8a**. Chromatographic purification (silica gel 40–60 μm) was performed by using CH_2_Cl_2_ as the eluent. Yield: 82%. mp. 165–167 °C (EtOAc/*n*-pentane). ^1^H-NMR (600 MHz, CDCl_3_) δ (ppm) 1.00 (d, 6H, *J* = 6.6 Hz, isopentyl (C*H_3_*)*_2_*CH), 1.77 (m, 2H, isopentyl C*H_2_*CH_2_O), 1.86 (m, 1H, isopentyl CH), 4.15 (t, 2H, *J* = 7.3 Hz, isopentyl CH_2_O), 6.03 (s, 1H, H-5), 6.77 (brs, 1H, D_2_O exch., NH), 6.96 (m, 1H, 3-fluorophenyl H-4), 7.13 (t, 1H, *J* = 7.3 Hz, aniline H-4), 7.38–7.45 (m, 3H, 3-fluorophenyl H-5, aniline H-3, H-5), 7.52 (d, 2H, *J* = 7.0 Hz, aniline H-2, H-6), 8.01 (s, 1H, H-3), 8.10 (m, 1H, 3-fluorophenyl H-6), 8.19 (m, 1H, 3-fluorophenyl H-2). ^13^C NMR (151 MHz, CDCl_3_) δ (ppm) 22.69, 25.26, 37.63, 67.34, 87.04, 104.94, 108.23, 108.41, 112.18, 112.27, 116.19, 121.16, 123.67, 129.39, 130.14, 130.20, 132.88, 139.98, 141.32, 151.76, 157.60, 161.49, 162.32,163.93. HPLC purity at 254 nm, 98.70%. HR-MS (ESI) *m/z* calculated for C_23_H_24_FN_4_O [M+H]^+^: 391.1929; found 391.1930.

4-(2-Bromoethoxy)-1-(3-fluorophenyl)-*N*-phenyl-*1H*-pyrazolo[3,4-*b*]pyridin-6-amine **(8f)**

This derivative was prepared by a method analogous to that described for the synthesis of **8a**. The reaction was completed after 6 h. Chromatographic purification (silica gel 40–60 μm) was performed by using a mixture of cyclohexane/CH_2_Cl_2_/THF (5/5/1, *v*/*v*) as the eluent. Yield: 80%. White solid, mp. 153–155 °C (EtOAc/*n*-pentane). ^1^H-NMR (600 MHz, acetone-*d_6_*) δ (ppm) 3.88 (d, 2H, *J* = 7.0 Hz, bromoethyl C*H_2_*CH_2_O), 4.58 (d, 2H, *J* = 7.0 Hz, bromoethyl CH_2_C*H_2_*O), 6.29 (s, 1H, H-5), 7.03–7.06 (m, 2H, 3-fluorophenyl H-4, aniline H-4), 7.37 (t, 2H, *J* = 7.3 Hz, aniline H-3, H-5), 7.55 (m, 1H, 3-fluorophenyl H-5), 7.81 (d, 2H, *J* = 7.0 Hz, aniline H-2, H-6), 8.03 (s, 1H, H-3), 8.25 (m, 1H, 3-fluorophenyl H-6), 8.34 (m, 1H, 3-fluorophenyl H-2), 8.69 (brs, 1H, D_2_O exch., NH). ^13^C NMR (151 MHz, acetone-*d_6_*) δ (ppm) 29.70 (overlapping with solvent), 69.24, 89.59, 104.96, 107.89, 108.07, 112.30, 112.44, 116.50, 120.60, 122.99, 129.58, 131.21, 131.27, 133.15, 141.81, 142.52, 142.59, 152.79, 158.64, 160.67, 162.99, 164.60. HPLC purity at 254 nm, 97.23%. HR-MS (ESI) *m/z* calculated for C_20_H_17_ BrFN_4_O [M+H]^+^: 427.0564; found 427.0560.

*N*,1-Diphenyl-4-[2-(phenylamino)ethoxy]-*1H*-pyrazolo[3,4-*b*]pyridin-6-amine **(9a)**

A solution of the bromide **8c** (40 mg, 0.098 mmol) and aniline (39 μL, 0.43 mmol) in EtOH (10 mL) was refluxed for 10 h. The solvent was vacuum-evaporated, and the residue was purified by flash column chromatography (silica gel 40–60 μm); a mixture of cyclohexane/CH_2_Cl_2_/THF (7/3/1 to 5/5/1, *v*/*v*) was used as the eluent, resulting in pure **9a** as an amorphous pale white solid. Yield 98%.^1^H-NMR (600 MHz, DMSO-*d_6_*) δ (ppm) 3.58 (m, 2H, C*H_2_*CH_2_O), 4.35 (t, 2H, *J* = 5.0 Hz, CH_2_C*H_2_*O), 5.84 (t, 1H, *J* = 5.7 Hz, D_2_O exch., NH-ethoxy), 6.29 (s, 1H, H-5), 6.57 (t, 1H, *J* = 7.3 Hz, phenylaminoethoxy H-4), 6.70 (d, 2H, *J* = 8.0 Hz, phenylaminoethoxy H-2, H-6), 6.98 (t, 1H, *J* = 7.3 Hz, aniline H-4), 7.11 (t, 2H, *J* = 7.6 Hz, phenylaminoethoxy H-3, H-5), 7.30–7.34 (m, 3H, phenyl H-4, aniline H-3, H-5), 7.55 (t, 2H, *J* = 7.6 Hz, phenyl H-3, H-5), 7.79 (d, 2H, *J* = 8.0 Hz, aniline H-2, H-6), 8.08 (s, 1H, H-3), 8.24 (d, 2H, *J* = 8.0 Hz, phenyl H-2, H-6), 9.42 (brs, 1H, D_2_O exch., NH-aniline). ^13^C NMR (151 MHz, DMSO-*d_6_*) δ (ppm) 41.83, 67.35, 88.70, 103.36, 112.23, 116.00, 118.78, 120.24, 121.34, 125.45, 128.58, 128.90, 132.19, 139.43, 140.90, 148.49, 151.06, 157.22, 159.64. HPLC purity at 254 nm, 96.72%. HR-MS (ESI) *m/z* calculated for C_26_H_24_N_5_O [M+H]^+^: 422.1975; found 422.1958.

*N*,1-Diphenyl-4-(2-(4-methylpiperazin-1-yl)ethoxy)-*1H*-pyrazolo[3,4-*b*]pyridin-6-amine **(9b)**

A solution of the bromide **8c** (50 mg, 0.12 mmol) and 4-methylpiperazine (109 μL, 0.98 mmol) in DMF (2.5 mL) was stirred at room temperature for 48 h. Upon completion of the reaction, the mixture was poured into cold water, the precipitate was filtered, washed with water, and air-dried. Chromatographic purification (silica gel 40–60 μm) was performed by using a mixture of CH_2_Cl_2_/MeOH (100/0.5 to 100/10, *v*/*v*) as the eluent. Yield: 65%. White solid, mp. 164–165 °C (EtOAc/*n*-pentane). ^1^H-NMR (600 MHz, CDCl_3_) δ (ppm) 2.30 (s, 3H, CH_3_), 2.48 (brs, 4H, piperazine H-3, H-5), 2.66 (brs, 4H, piperazine H-2, H-6), 2.90 (t, 2H, *J* = 7.4 Hz, C*H_2_*CH_2_O), 4.26 (t, 2H, *J* = 7.3 Hz, CH_2_C*H_2_*O), 6.04 (s, 1H, H-5), 6.68 (brs, 1H, D_2_O exch., NH), 7.10 (t, 1H, *J* = 7.3 Hz, aniline H-4), 7.27 (m, 1H, phenyl H-4, overlapping with solvent), 7.36 (t, 2H, *J* = 7.3 Hz, aniline H-3, H-5), 7.47–7.52 (m, 4H, phenyl H-3, H-5, aniline H-2, H-6), 8.01 (s, 1H, H-3), 8.26 (d, 2H, *J* = 7.0Hz, phenyl H-2, H-6). ^13^C NMR (151 MHz, CDCl_3_) δ (ppm) 46.16, 53.82, 55.23, 56.82, 66.97, 87.05, 104.66, 121.01, 121.27, 123.43, 125.71, 128.97, 129.36, 132.31, 140.01, 140.27, 151.91, 157.56, 160.95. HPLC purity at 254 nm, 97.27%. HR-MS (ESI) *m/z* calculated for C_25_H_29_N_6_O [M+H]^+^: 429.2397; found 429.2400.

4-(2-(Cyclohexylamino)ethoxy)-*N,*1-diphenyl-*1H*-pyrazolo[3,4-*b*]pyridin-6-amine **(9c)**

This derivative was prepared by a method analogous to that described for the synthesis of **9b**. Chromatographic purification (silica gel 40–60 μm) was performed by using a mixture of CH_2_Cl_2_/MeOH (100/0.5 to 100/15, *v*/*v*) as the eluent. Yield: 72%. White solid, mp. 143–145 °C (EtOAc/*n*-pentane). ^1^H-NMR (600 MHz, acetone-*d_6_*) δ (ppm) 1.08–1.22 (m, 3H, cyclohexyl H), 1.25–1.32 (m, 2H, cyclohexyl H), 1.58–1.60 (m, 1H, cyclohexyl H), 1.71–1.74 (m, 2H, cyclohexyl H), 1.91–1.93 (m, 2H, cyclohexyl H), 2.53–2.57 (m, 1H, cyclohexyl H), 3.11 (t, 2H, *J* = 5.0 Hz, C*H_2_*CH_2_O), 4.28 (t, 2H, *J* = 5.0 Hz, CH_2_C*H_2_*O), 6.29 (s, 1H, H-5), 7.00 (t, 1H, *J* = 7.3 Hz, aniline H-4), 7.30 (t, 1H, *J* = 7.3 Hz, phenyl H-4), 7.35 (t, 2H, *J* = 7.3 Hz, aniline H-3, H-5), 7.54 (t, 2H, *J* = 7.3 Hz, phenyl H-3, H-5), 7.84–7.86 (m, 2H, aniline H-2, H-6), 8.03 (s, 1H, H-3), 8.38 (d, 2H, *J* = 7.0 Hz, phenyl H-2, H-6), 8.64 (brs, 1H, D_2_O exch., NH-aniline). ^13^C NMR (151 MHz, acetone-*d_6_*) δ (ppm) 25.59, 26.99, 34.28, 46.05, 57.21, 69.81, 89.24, 104.98, 120.22, 121.37, 122.56, 126.12, 129.52, 129.63, 132.73, 141.17, 142.18, 152.53, 158.53, 161.42. HPLC purity at 254 nm, 98.65%. HR-MS (ESI) *m/z* calculated for C_26_H_30_N_5_O [M+H]^+^: 428.2445; found 428.2452.

1-(3-Fluorophenyl)*-N-*phenyl-4-(2-(phenylamino)ethoxy)-*1H-*pyrazolo[3,4-*b]*pyridin-6-amine **(9d)**

This derivative was prepared by a method analogous to that described for the synthesis of **9a**. Chromatographic purification (silica gel 40–60 μm) was performed by using a mixture of cyclohexane/EtOAc/CH_2_Cl_2_(9/1/1 to 7/3/1, *v*/*v*/*v*) as the eluent. Yield 73%. White solid, mp. 173–175 °C (EtOAc/*n*-pentane). ^1^H-NMR (600 MHz, CDCl_3_) δ (ppm) 3.67 (t, 2H, *J* = 5.0 Hz, C*H_2_*CH_2_O), 4.32 (t, 2H, *J* = 5.0 Hz, CH_2_C*H_2_*O), 6.02 (s, 1H, H-5), 6.65 (brs, 1H, D_2_O exch., NH), 6.70 (d, 2H, *J* = 7.5 Hz, phenylaminoethoxy H-2, H-6), 6.78 (t, 1H, *J* = 7.3 Hz, phenylaminoethoxy H-4), 6.96 (m, 1H, 3-fluorophenyl H-4), 7.13 (t, 1H, *J* = 7.3 Hz, aniline H-4), 7.23 (t, 2H, *J* = 7.3 Hz, phenylaminoethoxy H-3, H-5), 7.37–7.44 (m, 3H, 3-fluorophenyl H-5, aniline H-3, H-5), 7.51 (d, 2H, *J* = 7.1 Hz, aniline H-2, H-6), 8.02 (s, 1H, H-3), 8.10 (m, 1H, 3-fluorophenyl H-6), 8.20 (m, 1H, 3-fluorophenyl H-2). ^13^C NMR (151 MHz, CDCl_3_) δ (ppm) 43.11, 67.36, 87.30, 104.66, 108.13, 108.31, 112.09, 112.23, 113.34, 116.10, 118.39, 121.09, 123.67, 129.39, 129.60, 130.11, 130.17, 132.64, 139.95, 141.35, 141.42, 147.59, 152.05, 157.61, 160.85, 162.30, 163.92. HPLC purity at 254 nm, 95.66%. HR-MS (ESI) *m/z* calculated for C_26_H_23_FN_5_O [M+H]^+^: 440.1882; found 440.1881.

1-(3-Fluorophenyl)-4-[2-(4-methylpiperazin-1-yl)ethoxy]-*N*-phenyl-*1H*-pyrazolo[3,4-*b*]pyridin-6-amine **(9e)**

This derivative was prepared by a method analogous to that described for the synthesis of **9b**. Chromatographic purification (silica gel 40–60 μm) was performed by using a mixture of CH_2_Cl_2_/MeOH (100/8 to 100/16, *v*/*v*) as the eluent. Yield: 85%. Amorphous solid. ^1^H-NMR (600 MHz, DMSO-*d_6_*) δ (ppm) 2.26 (s, 3H, CH_3_), 2.50–2.59 (brs, 8H, piperazine), 2.84 (t, 2H, *J* = 5.0 Hz, C*H_2_*CH_2_O), 4.31 (t, 2H, *J* = 5.0 Hz, CH_2_C*H_2_*O), 6.31 (s, 1H, H-5), 7.01 (t, 1H, *J* = 7.0 Hz, aniline H-4), 7.13 (m, 1H, 3-fluorophenyl H-4), 7.34 (t, 2H, *J* = 7.5 Hz, aniline H-3, H-5), 7.57 (m, 1H, 3-fluorophenyl H-5), 7.77 (d, 2H, *J* = 8.0 Hz, aniline H-2, H-6), 8.08–8.11 (m, 2H, H-3, 3-fluorophenyl H-6), 8.26 (m, 1H, 3-fluorophenyl H-2), 9.55 (brs, 1H, D_2_O exch., NH). ^13^C NMR (151 MHz, DMSO-*d_6_*) δ (ppm) 45.09, 52.40, 54.33, 55.75, 66.42, 88.99, 103.60, 106.69, 106.87, 111.71, 111.84, 115.47, 119.13, 121.66, 128.62, 130.73, 130.79, 132.76, 140.75, 140.87, 140.95, 151.32, 157.43, 159.55, 161.45, 163.06. HPLC purity at 254 nm, 99.58%. HR-MS (ESI) *m/z* calculated for C_25_H_28_FN_6_O [M+H]^+^: 447.2303; found 447.2300.

4-[2-(Cyclohexylamino)ethoxy]-1-(3-fluorophenyl)-*N*-phenyl-*1H*-pyrazolo[3,4*-b*]pyridin-6-amine **(9f)**

This derivative was prepared by a method analogous to that described for the synthesis of **9b**. Chromatographic purification (silica gel 40–60 μm) was performed by using CH_2_Cl_2_/MeOH as the eluent (100/8 to 100/10, *v*/*v*). Yield: 85%. White solid, mp. 280–281 °C (EtOH/diethyl ether). ^1^H-NMR (600 MHz, DMSO-*d_6_*) δ (ppm) 1.07–1.16 (m, 3H, cyclohexyl H), 1.20–1.26 (m, 2H, cyclohexyl H), 1.56–1.58 (m, 1H, cyclohexyl H), 1.69–1.71 (m, 2H, cyclohexyl H), 1.88–1.90 (m, 2H, cyclohexyl H), 2.52–2.54 (m, 1H, cyclohexyl H, overlapping with solvent), 3.08 (t, 2H, *J* = 5.5 Hz, C*H_2_*CH_2_O), 4.25 (t, 2H, *J* = 5.5 Hz, CH_2_C*H_2_*O), 6.29 (s, 1H, H-5), 7.01 (t, 1H, *J* = 7.3 Hz, aniline H-4), 7.13 (m, 1H, 3-fluorophenyl H-4), 7.34 (t, 2H, *J* = 7.3 Hz, aniline H-3, H-5), 7.57 (m, 1H, 3-fluorophenyl H-5), 7.76 (d, 2H, *J* = 7.0 Hz, aniline H-2, H-6), 8.09 (m, 1H, 3-fluorophenyl H-6), 8.15 (s, 1H, H-3), 8.26 (m, 1H, 3-fluorophenyl H-2), 9.51 (brs, 1H, D_2_O exch., NH-aniline). ^13^C NMR (151 MHz, DMSO-*d_6_*) δ (ppm) 24.41, 25.73, 32.55, 44.52, 55.97, 68.28, 88.89, 103.62, 106.69, 106.87, 111.71, 111.85, 115.48, 119.18, 121.71, 128.63, 130.74, 130.80, 132.90, 140.74, 140.89, 140.96, 151.35, 157.46, 159.75, 161.46, 163.07. HPLC purity at 254 nm, 99.99%. HR-MS (ESI) *m/z* calculated for C_26_H_29_FN_5_O [M+H]^+^: 446.2351; found 446.2353.

1-Phenyl-6-(phenylamino)-*1H*-pyrazolo[3,4-*b*]pyridine-4-thiol **(10a)**

Lawesson’s reagent (448 mg, 1.11 mmoL) was added to a solution of the pyridinol **7a** (280 mg, 0.93 mmoL) in dioxane (15 mL), and the mixture was refluxed for 10 h. The solvent was then vacuum-evaporated, and the residue was purified by flash column chromatography (silica gel 40–60 μm). A mixture of CH_2_Cl_2_/EtOAc (100/0.5 to 100/40, *v*/*v*) was used as the eluent. Yield: 20%. Yellow solid, mp. 233–234 °C (EtOAc/*n*-pentane). ^1^H-NMR (600 MHz, CDCl_3_) δ (ppm) 6.79 (s, 1H, H-5), 6.85 (brs, 1H, D_2_O exch., NH), 7.01 (t, 1H, *J* = 7.3 Hz, aniline H-4), 7.22 (t, 2H, *J* = 7.0Hz, aniline H-3, H-5), 7.30 (t, 1H, *J* = 7.3 Hz, phenyl H-4), 7.43 (d, 2H, *J* = 7.0 Hz, aniline H-2, H-6), 7.49 (t, 2H, *J* = 7.0 Hz, phenyl H-3,H-5), 8.10 (s, 1H, H-3), 8.23 (d, 2H, *J* = 7.1 Hz, phenyl H-2, H-6). ^13^C NMR (151 MHz, CDCl_3_) δ (ppm) 101.85, 109.38, 120.63, 121.41, 123.63, 126.22, 129.09, 129.19, 132.02, 139.50, 139.54, 141.10, 149.87, 155.56. HPLC purity at 254 nm, 97.49%. HR-MS (ESI) *m/z* calculated for C_18_H_13_N_4_S [M-H]^-^: 317.0866; found 317.0850.

1-(3-Fluorophenyl)-6-(phenylamino)-*1H*-pyrazolo[3,4*-b*]pyridine-4-thiol **(10b)**

This derivative was prepared by a method analogous to that described for **10a**. Chromatographic purification (silica gel 40–60 μm) was performed by using a mixture of cyclohexane/EtOAc (75/25, *v*/*v*) as the eluent. Yield: 18%. White solid, mp. 210–211 °C (acetone). ^1^H-NMR (600 MHz, acetone-*d_6_*) δ (ppm) 7.02 (t, 1H, *J* = 7.3 Hz, aniline H-4), 7.05 (s, 1H, H-5), 7.12 (m, 1H, 3-fluorophenyl H-4), 7.31 (t, 2H, *J* = 7.3 Hz, aniline H-3, H-5), 7.61 (m, 1H, 3-fluorophenyl H-5), 7.73 (d, 2H, *J* = 7.0 Hz, aniline H-2, H-6), 8.24 (m, 1H, 3-fluorophenyl H-6), 8.30–8.32 (m, 2H, H-3, 3-fluorophenyl H-2), 9.02 (brs, 1H, D_2_O exch., NH). ^13^C NMR (151 MHz, acetone-*d_6_*) δ (ppm) 102.84, 107.21, 107.39, 108.71, 111.91, 112.05, 115.82, 119.69, 122.50, 128.62, 130.48, 130.54, 132.40, 140.12, 140.44, 141.07, 141.15, 149.82, 155.87, 162.01, 163.61. HPLC purity at 254 nm, 96.53%. HR-MS (ESI) *m/z* calculated for C_18_H_13_N_4_S [M-H]^−^: 335.0772; found 335.0762.

*N*,1-Diphenyl-*1H*-pyrazolo[3,4-*b*]pyridin-6-amine **(11a)**

Raney-Ni was added to a solution of the thiol **10a** (120 mg, 0.38 mmol) in EtOH (20 mL), and the mixture was refluxed for 4 h. The reaction mixture was filtered through a celite pad and washed with a mixture of CH_2_Cl_2_/MeOH (100/10, *v*/*v*). The filtrate was vacuum-evaporated and purified by flash column chromatography (silica gel 40–60 μm); a mixture of cyclohexane/EtOAc (95/5 to 80/20, *v*/*v*) was used as the eluent, resulting in pure **11a** (33 mg, 20%) as an amorphous solid. ^1^H-NMR (600 MHz, CDCl_3_) δ (ppm) 6.64 (d, 1H, *J* = 8.0 Hz, H-5), 6.82 (brs, 1H, D_2_O exch., NH), 7.12 (t, 1H, *J* = 7.3 Hz, aniline H-4), 7.29 (t, 1H, *J* = 7.3 Hz, phenyl H-4), 7.37 (t, 2H, *J* = 7.3 Hz, aniline H-3, H-5), 7.51 (t, 2H, *J* = 7.3 Hz, phenyl H-3, H-5), 7.60 (d, 2H, *J* = 7.0 Hz, aniline H-2, H-6), 7.84 (d, 1H, *J* = 8.0 Hz, H-4), 8.00 (s, 1H, H-3), 8.30 (d, 2H, *J* = 7.1 Hz, phenyl H-2, H-6). ^13^C NMR (151 MHz, CDCl_3_) δ (ppm) 106.73, 111.15, 120.52, 121.14, 123.26, 125.71, 128.99, 129.23, 131.50, 134.32, 139.88, 140.03, 149.85, 155.27. HPLC purity at 254 nm, 95.31%. HR-MS (ESI) *m/z* calculated for C_18_H_15_N_4_ [M+H]^+^: 287.1291; found 287.1289.

1-(3-Fluorophenyl)-*N*-phenyl-*1H*-pyrazolo[3,4*-b*]pyridin-6-amine **(11b)**

This derivative was prepared by a method analogous to that described for the synthesis of **11a**. Chromatographic purification (silica gel 40–60 μm) was performed by using CHCl_3_ as the eluent. Yield: 18%. Amorphous solid. ^1^H-NMR (600 MHz, CDCl_3_) δ (ppm) 6.60–6.62 (m, 1H, H-5), 6.85 (brs, 1H, D_2_O exch., NH) 6.97 (m, 1H, 3-fluorophenyl H-4), 7.13 (t, 1H, *J* = 7.3 Hz, aniline H-4), 7.38–7.45 (m, 3H, aniline H-3, H-5, 3-fluorophenyl H-5), 7.58 (d, 2H, *J* = 8.0 Hz, aniline H-2, H-6), 7.79–7.81 (m, 1H, H-4), 7.97 (s, 1H, H-3), 8.14 (m, 1H, 3-fluorophenyl H-6), 8.25 (m, 1H, 3-fluorophenyl H-2). ^13^C NMR (151 MHz, CDCl_3_) δ (ppm) 106.95, 107.93, 108.11, 111.25, 111.98, 112.12, 115.91, 115.93, 120.65, 123.47, 129.24, 130.10, 130.16, 131.52, 134.75, 139.73, 141.22, 141.29, 149.98, 155.34, 162.25, 163.87. HPLC purity at 254 nm, 97.14%. HR-MS (ESI) *m/z* calculated for C_18_H_14_FN_4_ [M+H]^+^: 305.1197; found 305.1196.

### 2.2. In Vitro Cytotoxicity (MTT) Assays

The human HCT116 colon cancer cell line and PC-3 prostate cancer cell line were obtained from the American Type Cell Culture (ATCC, Bethesda, MD, USA). HCT116 and PC-3 cell lines were grown at 37 °C in 5% CO_2_ using Roswell Park Memorial Institute 1640 medium (RPMI 1640) and Dulbecco’s modified Eagle’s medium F/12 (DMEM/F-12) containing 10% fetal bovine serum (FBS). MEFs and KPC cells were cultured in a 5% CO_2,_ with DMEM medium containing 10% fetal bovine serum (FBS), 100 U/mL Penicillin and 100 mg/mL Streptomycin at 37 °C. EO771 cells were purchased from CH3 BioSystems and cultured in complete RPMI 1640 medium supplemented with 10% fetal calf serum, 100 U/mL Penicillin and 100 mg/mL Streptomycin. To test the inhibitory activities of compounds using a cell-based assay, MTT assays were performed for cell viability. Briefly, in 96-well plates, HCT116 cells were plated at a density of 1500 per well, PC-3 cells were plated at a density of 750 per well; KPC and MEFs were plated at a density of 3000 cells per well; and EO771 were seeded at a density of 2500 cells per well. The differences in the initial seeding cell numbers reflect the differences in the doubling time of cells, given that we sought to have wells that were close to but not confluent at the end of the experiment. After 24 h, cells were treated with the indicated compounds in a dose-dependent manner for 72 h and 96 h. Viable cell numbers were determined by tetrazolium conversion to its formazan dye. All the experiments were performed three times, and the tested concentrations in each experiment were evaluated in quadruplicated wells. Mouse embryonic fibroblasts (MEFs) and KPC pancreatic cells (Kras(mut); Pdx Cre) are cell lines derived from our C57/Bl6 mice and the double transgenic mouse pancreatic cancer model, respectively [21,22,23].

### 2.3. Mouse Models

All studies were approved by the National Hellenic Research Foundation Animal Care and Use Committee. The study protocol was approved by the local ethics committee (Athens Prefecture Veterinarian Service; (431956/17-05-2022.) Animal care was provided in accordance with the procedures outlined in the “Guide for Care and Use of Laboratory Animals (National Research Council; 1996; National Academy Press; Washington, DC, USA). C57BL/6 female mice were purchased from Jackson Laboratory. Mice at 6–8 weeks of age were randomly assigned to treatment or control groups. As indicated by the performed power analysis (clinical calculator/clincalc.com), 5 mice per group were used in each of the in vivo experiments. We repeated the in vivo experiments three times and found similar results. In the results section, we present our data from one experiment. The EO771 were cultured in complete RPMI 1640 medium supplemented with 10% fetal calf serum, 50 mM 2-mercaptoethanol, 100 U/mL Penicillin, and 100 mg/mL Streptomycin. Murine EO771 (5 × 10^5^) were orthotopically inoculated at the fourth mammary fat pad of 6–8 weeks old mice. The cells were resuspended in PBS. Matrigel (Corning Inc., Corning, NY, USA) was added at 1:3 dilution to facilitate the inoculation process. Matrigel, an extract of basement membrane proteins, was used as a cell carrier medium for cell transplantation studies by forming a 3D gel at 37 °C that facilitated the inoculation. Tumor size was measured using a digital caliper, and tumor volume (mm^3^) was calculated by the following equation: L*W*H*π/6. The mice were monitored on a daily basis for any signs of discomfort and the orthotopic tumors were also monitored routinely for any signs of ulcerations or any other type of wounds. The mice were also weighted once a week. Per our guidelines, any mouse found with an ulcerated wound, 30% weight loss, or visual signs of discomfort (slow reflexes, not walking normally, hunched back, or rough coat) was immediately excluded from the experiment and euthanized. Despite the strict rules, no mice were excluded from our experiments. All the treatments were well tolerated by the mice, showing no signs of toxicity.

### 2.4. Treatment of EO771 Tumor-Bearing Mice

Treatment was initiated when tumors reached ~20 mm^3^. Animals were treated with **9b**, **9c** and **9e** peritumorally by administration of 100 μg of the aforementioned compounds in Matrigel (Corning Inc.), used in 1:4 dilution, every 4 days. The mice were sacrificed when the primary tumor reached a 2 cm diameter, or at any other humane endpoints as listed in the ACUC-approved animal protocol, such as 20% weight loss or acute morbidity.

### 2.5. Histology and Immunohistochemistry Staining

Tissue samples, including tumors, were fixed in 10% neutral buffered formalin (NBF, Sigma), then routinely processed and paraffin embedded. Tumor and lung sections were dewaxed and rehydrated, then stained with hematoxylin and eosin (H&E). For immunohistochemistry, sections were antigen-retrieved with the heat-induced or enzymatic method. Peroxidase activity was blocked using 1.5% hydrogen peroxide. Sections were blocked with different blocking protocols, depending on the antibody. Staining was performed using the following anti–mouse antibodies: anti-Ki67 (Cell Signaling, 9449) (1:1000 dilution) and anti-Caspase 3 (Cell Signaling, 9661) (1:800 dilution). A polymer-based detection kit, which consists of horseradish peroxidase-conjugated polymers was used for detection. To determine proliferation indices, Ki67-positive and Ki67-negative cells were counted using ImageJ software (US National Institutes of Health) in 8–10 representative fields of all tumors (on average, ~3000 nuclei were counted per specimen). A similar approach was followed to evaluate the % percentage of apoptotic cells.

### 2.6. Statistical Analysis

Statistical analyses and graph generation were performed with GraphPad Prism 9.2.0 (San Diego, CA, USA). Tumor areas were plotted as mean ± standard error of the mean (SEM) for each data point, and tumor growth curves were compared using mixed effects ANOVA. Differences were evaluated by 1-way ANOVA or unpaired parametric Student’s *t* test. The *p*-values were calculated for multiple comparisons using Tukey’s multiple comparisons test.

## 3. Results and Discussion

### 3.1. Synthesis of the Novel Compounds

Upon a literature search for the optimum way to accomplish the synthesis of pyrazolo[3,4-b]pyridines [24,25], we have decided to prepare the target compounds by using commercial phenylhydrazine (**1a**) or 3-fluorophenylhydrazine (**1b**) as starting materials (Figure 1). Each hydrazine was converted to the corresponding pyrazole-4-carbonitriles (**2a**,**b**) [26,27], and then, upon hydrolysis and subsequent decarboxylation, converted to the pyrazoles **3a**,**b** following a previously reported methodology [20,28]. The above mentioned pyrazoles were then treated with the Meldrum’s acid dimethythiomethylene-derivative **4** [29] to provide the intermediate dioxanediones **5a**,**b**. The remaining methylthio group of compounds **5a**,**b** were then displaced upon reaction with aniline, and the derived dioxanediones **6a**,**b** were subjected to thermal cyclization and converted to the pyrazolopyridinols **7a**,**b**.

The pyridinones **7a**,**b** were treated with K_2_CO_3_ and suitable bromides to provide the corresponding ethers through their enolates **8a**–**f**. The unambiguous confirmation for the O-alkylation was evidenced from 2D NMR data. Thus, from the HMBC spectrum, a cross-peak between the most downfield ethyloxy side chain methylene with C-4 was obvious, and additionally, data from the nOe spectrum showed that the above-mentioned methylene correlates to both H-5 and H-3 as well. Additionally, the bromides **8c**,**f** were used for the introduction of selected amines, namely aniline, 4-methylpiperazine, or cyclohexylamine, which resulted in the aminoderivatives **9a**–**f**. Along with the aim of expanding the structural diversity of this scaffold, compounds **7a**,**b** were allowed to react with Lawesson’s reagent, and the resulting thiols **10a**,**b** were reduced by using Raney nickel as the catalyst to provide the corresponding 1,6-disubstituted pyrazolopyridines **11a**,**b**. An alternative pathway for the synthesis of a **7a** modified analogue is also presented in the Appendix A in Appendix A.

### 3.2. Evaluation of the Antiproliferative Efficacy of the New pyrazolo[3,4-b]pyridine Derivatives In Vitro

The cytotoxic activity of all target compounds, **7a**,**b**, **8a**–**f**, **9a**–**f**, **10a**,**b,** and **11a**,**b,** was initially evaluated against the prostatic PC-3 and the colon HCT116 human cancer cell lines. This screening provided very interesting information concerning the structural requirements for activity. The majority of the compounds did not exhibit significant activity; they possessed IC_50_ values greater than 10 µM, with the remarkable exception of two couples of analogously substituted derivatives, specifically, **9b**,**c** and **9e**,**f**. The latter were endowed with low µM activity in the IC_50_ range of 0.75–4.55 µM (Table 1). All four compounds are 4-alkylaminoethoxy derivatives, providing an indication that the presence of a 4-arylaminoether, a 4-arylalkyl or 4-alkyl ether do not encourage activity. This is also true in the case of pyrazolopyridinones **7a**,**b**, the corresponding thiones **10a**,**b,** and the 4-unsubstituted analogues **11a**,**b**. These active analogues were subsequently tested against two additional cancer cell lines and one non-cancerous murine cell line. The first murine cancer cell line, EO771, is a breast cancer cell line, syngeneic to C57/Bl6; the second cell line, KPC, is a pancreas cancer cell line, syngeneic to C57/Bl6. The non-cancerous murine cell line consists of mouse embryonic fibroblasts (MEFs) and is also derived from C57/Bl6 mice.

Cell viability of the four derivatives against normal mouse embryonic fibroblasts (MEFs) was also examined. Three compounds were found non-toxic (**9b**, **9c** and **9e**, presenting IC_50_ >200 µM), and the fourth showed severe cytotoxicity with an IC_50_ = 11.2 µM. Figure 2 depicts the comparison of the IC_50_ curves of the selected **9b** (Figure 2A), **9c** (Figure 2B), **9e** (Figure 2C), and **9f** (Figure 2D) in mouse embryonic cells and EO771 mouse breast cancer cell. The difference in the cell viability between those cell lines indicate possible anti-breast cancer efficacy of these compounds and minimal off target effects. It is noteworthy that those compounds were not as effective in reducing the cell proliferation of the KPC mouse pancreatic cell line, suggesting possible breast cancer specificity. Based on our in vitro data, we decided to investigate the efficacy of **9b**, **9c,** and **9e** in vivo in our tumor bearing EO771 mouse model. Only **9f** was excluded from the subsequent in vivo experiments because of its low IC_50_ values on the MEFs, so that potential increased toxicity or lethality to the experimental animals could be avoided.

### 3.3. Evaluation of the Anti-Cancer Efficacy of the Most Active Analogues In Vivo

We next sought to investigate the efficacy of the novel compounds in mouse syngeneic models of cancer, enabling us to monitor the effects of the treatment not only to the tumor growth itself, but also to the immune system of the mouse, which better simulated the interactions we often monitor in clinical settings. For subsequent in vivo analysis, we chose EO771 cells as they showed very good IC_50_ values and specificity in comparison to the pancreatic cell line or the non-cancerous fibroblasts. We thus established mouse tumors by using EO771 cells and orthotopically inoculated into the mammary fat pad of the mouse. At 6–7 days after inoculating 5 × 10^5^ cells into the mammary fad pad, the mice develop palpable tumors; immediately after this observation, we initiated our treatment with the selected compounds. Based on our previous in vivo studies [21], and the fact that all the in vivo evaluated agents showed minimal toxicity on the mouse embryonic fibroblasts, we initiated a treatment protocol of intraperitoneal delivery of 100 μg every four days until day 26 post-inoculation. As presented in Figure 3, all the tested compounds resulted in a strong inhibition of tumor growth (Figure 3A). The compound-specific tumor growth curves revealed that **9b** was more potent in inhibiting tumor growth in comparison to the other analogues. (Figure 3B–D). The treatment schedule is depicted in Figure 3E.

### 3.4. Histological Analysis of the Tumors

The histology of the tumors revealed that all treatments affected both the proliferation of the cancer cells and their overall survival. Representative pictures of the Ki67 staining are depicted in Figure 4A. The proliferation index, measured by Ki67 immunohistochemistry (Figure 4B), showed that the control tumors had increased proliferation in contrast to the treated tumors (76.74% control vs. 41.95% **9b**, 43.47% **9c** and 41.08% **9e**). The reduced proliferation rates of all treated groups were statistically significant when compared to the untreated controls (*p* = 1.23 × 10^−12^, *p* = 3.3 × 10^12^ and *p* = 1.53 × 10^−13^ for **9b**, **9c** and **9e** respectively.

We next decided to analyze the apoptotic rate of cancer cells by using cleaved caspase-3 (Figure 5). For that purpose, we evaluated the staining in the periphery of the tumors rather than the staining on the central necrotic part. (Figure 5A). We observed an increase of the apoptotic cells in all the treated groups. The cleaved caspase-3 staining revealed a significant increase of the apoptotic cells (*p* = 1.32 × 10^−5^, *p* = 0.03, *p* = 0.002) in **9b**, **9c,** and **9e,** respectively. The mean percentages of apoptosis were 6.1% for the controls, 9.9% for **9b** treated, 7.68% for **9c,** and 8.37% for **9e** respectively. Most importantly, the **9b** treated tumors showed statistically significant increase in apoptosis in comparison to the other treated groups (*p* = 0.009 with **9c** and *p* = 0.049 with **9e**), which is also in line with our in vitro experiments.

## 4. Conclusions

In this work, a number of substituted pyrazolo[3,4-*b*]pyridines were prepared and tested for their antiproliferative activity. We decided not to alter the nature of the substituent in positions 1 and 6 of the scaffold; thus, we inserted 1-phenyl or 1-(3-fluorophenyl) group together with 6-phenylamino group in all target compounds. In total, we used a variety of 4-substituents and have studied their impact on the biological activity. Interestingly, the in vitro evaluation revealed that only 4-alkylaminoethoxy derivatives possessed strong cytotoxicity against the four cancer cell lines tested in the low µM range, while the normal cell line (normal mouse embryonic fibroblasts) remains practically unaffected. This finding prompted us to study the in vivo efficacy of the three most active analogues in a mouse breast cancer model. All compounds, and more profoundly, the 4-(4-methylpiperazin-1-yl)ethoxy derivative **9b**, presented a strong inhibition of tumor growth and induced apoptosis. It should be noted that these compounds do not show systemic toxicity and did not interfere with the immune system of the animals. More comprehensive structure–activity relationships, e.g., the necessity of the 1-phenyl (or fluorophenyl) substituent, as well as of the 6-phenylamino group, is currently under active investigation in our laboratories, with the aim of assisting in structure optimization and elucidating the molecular mode of action of this class of compounds.

## Data Availability

All data are available upon request.

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
