# Peer review of "Discovery of New 1,4,6-Trisubstituted-1H-pyrazolo[3,4-b]pyridines with Anti-Tumor Efficacy in Mouse Model of Breast Cancer"

_pharmaceutics, 2023, doi:10.3390/pharmaceutics15030787_

Round 1

Reviewer 1 Report

COMMENTS:

·         In the present study, the authors described the synthesis of some novel substituted pyrazolo[3,4-b]pyridine derivatives and investigated for their ability to inhibit proliferation. The brand-new products effectively inhibited tumor development and brought on apoptosis. It should be highlighted that these compounds do not exhibit systemic toxicity and do not affect the animals' immune systems. Both the writing and the experiments are convincing. The references section is relevant, and the data give the conclusions significant support. Taking into account the following requires some thought:

·         Every compound title that is written in the experimental part must always start with a capital letter. For ex.

·         [2,2-dimethyl-5-{(methylthio)[(1-phenyl-1H-pyrazol-5-yl)amino]methylene}-1,3-di-oxane-4,6-dione (5a)] should be corrected to [2,2-Dimethyl-5-{(methylthio)[(1-phenyl-1H-pyrazol-5-yl)amino]methylene}-1,3-di-oxane-4,6-dione (5a)] and all the other compounds carry on in the same way.

·         The molecular formula and molecular weight for each compound in the experimental section should be carefully revised.

·         Corrections must be made to the names of the compounds (7a) and (7b).

·         Why is the "Na" atom present in the compound (8b) formula, [C23H24N4NaO]? and the molecular weight [395.1843] was used in the estimation?

·         The 1HNMR spectra of compounds (7a), (7b), (10a) and (10b) contain no signals referring to the (-OH) and (-SH) functions, respectively. So, the results of the spectroscopy for some derivatives need to be reconsidered.

·         I propose eliminating (Scheme 2) from the article because (Scheme 1) has a number of reactions while (Scheme 2) compounds have no biological impact. Moreover, the authors reported "our attempts to substitute the 6-methylthiogroup of 13 or even the comparable oxidised 6-methylsulfonylderivative 14 were not successful."

·         A molecular docking technique required to be included for the most active molecule being studied.

·         Verify that the manuscript is free of any typographical, spacing, or font writing errors.

Author Response

We thank the Reviewer for his/her fruitful comments, that helped in the amelioration of our article.

COMMENTS:

  • In the present study, the authors described the synthesis of some novel substituted pyrazolo[3,4-b]pyridine derivatives and investigated for their ability to inhibit proliferation. The brand-new products effectively inhibited tumor development and brought on apoptosis. It should be highlighted that these compounds do not exhibit systemic toxicity and do not affect the animals' immune systems. Both the writing and the experiments are convincing. The references section is relevant, and the data give the conclusions significant support. Taking into account the following requires some thought:

  • Every compound title that is written in the experimental part must always start with a capital letter. For ex. [2,2-dimethyl-5-{(methylthio)[(1-phenyl-1H-pyrazol-5-yl)amino]methylene}-1,3-di-oxane-4,6-dione (5a)] should be corrected to [2,2-Dimethyl-5-{(methylthio)[(1-phenyl-1H-pyrazol-5-yl)amino]methylene}-1,3-di-oxane-4,6-dione (5a)] and all the other compounds carry on in the same way.

Answer: We have corrected this point, all compounds in the titles of the experimental part, start with capital letter.

  • The molecular formula and molecular weight for each compound in the experimental section should be carefully revised.

Answer: We have revised the molecular formulas and weights, more precisely, we altered the presentation mode providing each molecular ion.

  • Corrections must be made to the names of the compounds (7a) and (7b).

Answer: These compounds have been renamed, as proposed by the Reviewer.

  • Why is the "Na" atom present in the compound (8b) formula, [C23H24N4NaO]? and the molecular weight [395.1843] was used in the estimation?

Answer: This atom reflects the mode that the MS spectrum was recorded (MS adduct ion).

  • The 1HNMR spectra of compounds (7a), (7b), (10a) and (10b) contain no signals referring to the (-OH) and (-SH) functions, respectively. So, the results of the spectroscopy for some derivatives need to be reconsidered.

Answer: All four compounds are enolates, in solution both the pyridinol and pyridinone or the corresponding thiol/thione forms are present due to tautomerism, consequently the OH or SH 1HNMR signal may not be recordable.

  • I propose eliminating (Scheme 2) from the article because (Scheme 1) has a number of reactions while (Scheme 2) compounds have no biological impact. Moreover, the authors reported "our attempts to substitute the 6-methylthiogroup of 13 or even the comparable oxidised 6-methylsulfonylderivative 14 were not successful."

Answer: We have included in our originally submitted manuscript Scheme 2, as an information concerning the chemistry of this class of derivatives (compounds 12-14), not having in mind their biological impact. It should be noted that compounds 12-14 have not been reported previously (new compounds) and they could provide some potentially important data to the scientific community (to scientists involved in chemistry research). Nevertheless, following the Reviewer’s suggestion we have omitted Scheme 2 from the revised manuscript and have included this, as Scheme SI1, together with the corresponding experimental data concerning compounds 12-14, in the Supporting Information section.

  • A molecular docking technique required to be included for the most active molecule being studied.

Answer: We thank the reviewer for this remark. In fact, this is our goal and we keep working hard in order to elucidate the exact mechanism of action of this type of compounds. Unfortunately, we have not discovered the molecular target yet. We have of course considered a number of kinases, but we have not identified anything specific for the moment.

During our efforts, we performed molecular docking using the PharmMapper server for the most active compound, 9b. PharmMapper server is a freely accessed web server designed to identify potential target candidates for the given small molecules (drugs, natural products or other newly discovered compounds with unidentified binding targets) using pharmacophore mapping approach. PharmMapper hosts a large, in-house repertoire of pharmacophore database (namely PharmTargetDB) annotated from all the targets information in TargetBank, BindingDB, DrugBank and potential drug target database, including over 7000 receptor-based pharmacophore models (covering over 1500 drug targets information). Among the 301 proteins which had the highest fit the top 10 are the following:

  1. Cytochrome b5. (Normalized fit 0.85)
  2. Glycerol-3-phosphate dehydrogenase [NAD+], glycosomal. (Normalized fit 0.84)
  3. Cadherin-2. (Normalized fit 0.84)
  4. Periplasmic [NiFe] hydrogenase small subunit. (Normalized fit 0.83)
  5. Methionine synthase reductase, mitochondrial. (Normalized fit 0.82)
  6. E3 SUMO-protein ligase RanBP2. (Normalized fit 0.82)
  7. Methylglyoxal synthase. (Normalized fit 0.82)
  8. DNA polymerase. (Normalized fit 0.81)
  9. Transcription elongation regulator 1. (Normalized fit 0.81)
  10. Type II DNA topoisomerase VI subunit A. (Normalized fit 0.80)

Our molecular docking results indicate possible multiple targets of the studied compound; therefore, we will continue exploring the possible unique target using thermal proteome profiling analysis. The top ten results from the molecular docking experiment indicate possible interaction either with the cellular metabolism pathways or with the DNA replication machinery, a fact which results in reduced proliferation and apoptosis, and is confirmed with our in vitro and in vivo data.

Overall, since we would not wish to speculate, we cannot perform substantial docking studies until we identify a specific protein or cellular function that will be responsible for the activity.

  • Verify that the manuscript is free of any typographical, spacing, or font writing errors.

Answer: We have tried our best to eliminate any typographical errors. We have highlighted the major points we have corrected in the revised version of our manuscript.

Reviewer 2 Report

Georgiou et al. developed new 1,4,6-trisubstituted-1H-pyrazolo[3,4-b]pyridines with anti-tumor efficacy in a mouse model of breast cancer. The study is exciting, and the compounds displayed potent anti-cancer activity.

I have some concerns which need to be addressed before the publication of this ms

1.      Designing of compounds should be explored.

2.      Authors should investigate the molecular target of these compounds.

3.      The mechanism of action of these derivatives should be investigated

4.      HPLC purity data should be included.

5.      Typo errors should be corrected in the whole ms. 

Author Response

We thank the Reviewer for his/her fruitful comments, that helped in the amelioration of our article.

Comments and Suggestions for Authors

Georgiou et al. developed new 1,4,6-trisubstituted-1H-pyrazolo[3,4-b]pyridines with anti-tumor efficacy in a mouse model of breast cancer. The study is exciting, and the compounds displayed potent anti-cancer activity.

I have some concerns which need to be addressed before the publication of this ms

  1. Designing of compounds should be explored.

Answer: We have modified the introduction (3rd paragraph), in order to provide more information on the way we have treated previous results of our group, so as to design the new compounds presented in this study. These additions are highlighted in the introductory part.

  1. Authors should investigate the molecular target of these compounds.

Answer: As we mentioned in our response to Reviewer No 1, we have not identified the molecular target yet. Our laboratories are currently working in the continuation of this project, synthesizing new structural analogues that could assist in the clarification of the biological target.

Please allow us to copy here our exact response to Reviewer No 1:

We thank the reviewer for this remark. In fact, this is our goal and we keep working hard in order to elucidate the exact mechanism of action of this type of compounds. Unfortunately, we have not discovered the molecular target yet. We have of course considered a number of kinases, but we have not identified anything specific for the moment.

During our efforts, we performed molecular docking using the PharmMapper server for the most active compound, 9b. PharmMapper server is a freely accessed web server designed to identify potential target candidates for the given small molecules (drugs, natural products or other newly discovered compounds with unidentified binding targets) using pharmacophore mapping approach. PharmMapper hosts a large, in-house repertoire of pharmacophore database (namely PharmTargetDB) annotated from all the targets information in TargetBank, BindingDB, DrugBank and potential drug target database, including over 7000 receptor-based pharmacophore models (covering over 1500 drug targets information). Among the 301 proteins which had the highest fit the top 10 are the following:

  1. Cytochrome b5. (Normalized fit 0.85)
  2. Glycerol-3-phosphate dehydrogenase [NAD+], glycosomal. (Normalized fit 0.84)
  3. Cadherin-2. (Normalized fit 0.84)
  4. Periplasmic [NiFe] hydrogenase small subunit. (Normalized fit 0.83)
  5. Methionine synthase reductase, mitochondrial. (Normalized fit 0.82)
  6. E3 SUMO-protein ligase RanBP2. (Normalized fit 0.82)
  7. Methylglyoxal synthase. (Normalized fit 0.82)
  8. DNA polymerase. (Normalized fit 0.81)
  9. Transcription elongation regulator 1. (Normalized fit 0.81)
  10. Type II DNA topoisomerase VI subunit A. (Normalized fit 0.80)

Our molecular docking results indicate possible multiple targets of the studied compound; therefore, we will continue exploring the possible unique target using thermal proteome profiling analysis. The top ten results from the molecular docking experiment indicate possible interaction either with the cellular metabolism pathways or with the DNA replication machinery, a fact which results in reduced proliferation and apoptosis, and is confirmed with our in vitro and in vivo data.

Overall, since we would not wish to speculate, we cannot perform substantial docking studies until we identify a specific protein or cellular function that will be responsible for the activity.

  1. The mechanism of action of these derivatives should be investigated

Answer: Please refer to our response in the previous comment (number 2).

  1. HPLC purity data should be included.

Answer: We have included HPLC purity data concerning all compounds tested in vitro, according to the Reviewer’s suggestion.

  1. Typo errors should be corrected in the whole ms.

Answer: We have tried our best to eliminate any typographical errors. We have highlighted the major points we have corrected in the revised version of our manuscript.

Reviewer 3 Report

1. The optimal Conclusion should include:
• A summary of your key findings.
• A highlight of your hypothesis, new concepts and innovations.

• Your vision for future work.

2.the number of repeat experiments underlying error estimates should be specified in each figure/table caption.

3. For the design and synthesis of 1,6-disubstituted pyrazolo[3,4-b]pyridine-4-ones, some current work could be considered, such as Polym. Chem., 2022, 13, 2351–2361; Chem. Commun., 2022, 58, 6653–6656; Org. Chem. Front., 2020,7, 3515-3520; New J. Chem., 2020, 44, 16265-16268; J. Org. Chem. 2019, 84, 14627−14635 and Org. Chem. Front., 2021, 8, 4554–4559

4.Ethical permit for animal experiments should be specified. Details for how these experiments were run must also be provided.

5.  MTT assay, IC50 of different drug group should be supplemented. 

6. Fig. 3b-d was suggested to change with tumor inhibition rate.

7. Standard deviations should be provided for FIG2.

8.MTT assay was conducted on EO771 cells and other cells. Justifications for selecting these three cells should be provided.

Author Response

We thank the Reviewer for his/her fruitful comments, that helped in the amelioration of our article.

Comments and Suggestions for Authors

  1. The optimal Conclusion should include:
  • A summary of your key findings.
  • A highlight of your hypothesis, new concepts and innovations.
  • Your vision for future work.

Answer: We would like to thank the Reviewer for his/her remarks. From our point of view, we think we have included in the conclusion all the key points that the Reviewer mention, albeit in a rather brief manner, since we prefer to avoid a possible speculation in any subject that needs to be further clarified.

2.the number of repeat experiments underlying error estimates should be specified in each figure/table caption.

Answer: We have included the requested information in each figure legend, following the Reviewer recommendations.

  1. For the design and synthesis of 1,6-disubstituted pyrazolo[3,4-b]pyridine-4-ones, some current work could be considered, such as Polym. Chem., 2022, 13, 2351–2361; Chem. Commun., 2022, 58, 6653–6656; Org. Chem. Front., 2020,7, 3515-3520; New J. Chem., 2020, 44, 16265-16268; J. Org. Chem. 2019, 84, 14627−14635 and Org. Chem. Front., 2021, 8, 4554–4559

Answer: We have incorporated information concerning the synthesis of pyrazolo[3,4-b]pyridine derivatives, precisely, Refs No 20 and 21. We have accordingly renumbered the following references of the manuscript.  

4.Ethical permit for animal experiments should be specified. Details for how these experiments were run must also be provided.

Answer: The ethical permit and the guidelines are specified in the materials and methods section in paragraph 4.3 mouse models, in lines 2-6. We have also added at the end of this paragraph details on the way the animals were treated (The mice were monitored on a daily basis for any signs of discomfort and the orthotopic tumors were also monitored routinely for any signs of ulcerations or any other type of wounds. Once a week the mice were also weighted.  Per our guidelines any mouse found with ulcerated wound or with 30% weight loss or with visual signs of discomfort (slow reflexes not walking normally, hunched back or with rough coat) is immediately excluded from the experiment and euthanized. Despite the strict rules we have never excluded any mouse from any of our experiments. All the treatments were well tolerated by the mice, showing no signs of toxicity).

  1. MTT assay, IC50 of different drug group should be supplemented.

Answer: The IC50 of DOXORUBICIN is provided in the last row of Table 1.

  1. Fig. 3b-d was suggested to change with tumor inhibition rate.

Answer: We tried to change the graphs according to the Reviewer’s suggestions using the following formula: %TGI = [(1-(Vt1/Vto)/(Cto/Ct1))/(1-Cto/Ct1)]*100. The new graphs did not provide any better information and were rather confusing due to the fact that they could not recapitulate the inhibition rate in the cases we had complete eradication of the tumors. Therefore, we suggest to keep the Figure 3d-b unchanged.

  1. Standard deviations should be provided for FIG2.

Answer: The standard deviations are provided in FIG2, but are not visible due to their small numbers. Please see the corresponding values in Table 1.

  1. MTT assay was conducted on EO771 cells and other cells. Justifications for selecting these three cells should be provided.

Answer: We decided initially to study any potential efficacy of the novel compounds in human cancer cell lines, therefore we selected HCT116, and PC-3, due to the fact that these cell lines were rapidly growing and thus compounds like pyrazolo[3,4-b]pyridine derivatives, might interfere with their rapid DNA replication. Concerning the in vivo experiments, we have added a comment in paragraph 2.3 (highlighted), following the Reviewers’ suggestion.

Following the Editor’s suggestion, we provided information concerning the origin of the human HCT116 colon cancer cell line and PC-3 prostate cancer cell line. This is highlighted in the experimental part.

Round 2

Reviewer 1 Report

There are still several writing faults in all the manuscript. For ex.

Figure 3: It should be changed to, and brackets should be added (Figure 3)

Fig. 3E: It should be changed to and put in brackets (Figure 3E)

Figure 4a: It should be changed to and put in brackets (Figure 4a)